# Describing Independent Eating Occasions among Low-Income Adolescents

**DOI:** 10.3390/ijerph17030981

**Published:** 2020-02-05

**Authors:** Jinan Banna, Rickelle Richards, Blake Jones, Alex Kojo Anderson, Marla Reicks, Mary Cluskey, Carolyn Gunther, Nobuko Kay Hongu, Karina Lora, Scottie Misner, Lillie Monroe-Lord, Glade Topham, Siew Sun Wong, Eunjung Lim

**Affiliations:** 1Department of Human Nutrition, Food and Animal Sciences, University of Hawaii-Manoa, Honolulu, HI 96822, USA; 2Department of Nutrition, Dietetics & Food Science, Brigham Young University, Provo, UT 84602, USA; rickelle_richards@byu.edu; 3Department of Psychology, Brigham Young University, Provo, UT 84602, USA; blake.jones@byu.edu; 4Department of Foods and Nutrition, University of Georgia, Athens, GA 30602, USA; fianko@uga.edu; 5Department of Food Science and Nutrition, University of Minnesota, St. Paul, MN 55108, USA; mreicks@umn.edu; 6School of Biological and Population Health Sciences, Nutrition, Oregon State University, Corvallis, OR 97331, USA; cluskeym@oregonstate.edu (M.C.); siewsun.wong@oregonstate.edu (S.S.W.); 7Department of Human Sciences, Ohio State University, Human Nutrition Program, Columbus, OH 43210, USA; Gunther.22@osu.edu; 8Department of Nutritional Sciences, University of Arizona, Tucson, AZ 85721, USA; hongu@email.arizona.edu (N.K.H.); misner@email.arizona.edu (S.M.); 9Department of Exercise and Nutrition Sciences, George Washington University, Washington, DC 20052, USA; klora@email.gwu.edu; 10Center for Nutrition, Diet and Health, University of the District of Columbia, Washington, DC 20008, USA; lmonroelord@udc.edu; 11Couple and Family Therapy Program, Kansas State University, Manhattan, KS 66506, USA; gtopham@ksu.edu; 12Department of Quantitative Health Sciences, University of Hawaii-Manoa, Honolulu, HI 96813, USA; lime@hawaii.edu

**Keywords:** adolescents, low-income, diet quality, independent eating occasions, parents

## Abstract

The purpose of this formative, cross-sectional study was to describe independent eating occasions (iEOs) among a convenience sample of low-income early adolescents (10–13 years, n = 46) in 10 U.S. states, including environmental context, foods selected and reasons for selection, and parental rules about foods consumed. Participants took pictures of all foods consumed over 24 h and participated in semi-structured interviews to describe the context of each eating occasion using the pictures as a guide. Responses based on a total of 304 eating occasions were coded to classify foods based on United States Department of Agriculture (USDA) MyPlate food groups and to characterize parental rules and reasons for food selection. Average age was 10.9 ± 1.1 years and 60% were female. Approximately 58% of eating occasions were classified as iEOs with approximately 65% as snacks. Most iEOs took place at home. Foods frequently consumed during iEOs were from the sweets, total fruit, dairy, and whole fruit food categories. Primary parental rules for iEOs focused on avoiding certain foods and not eating too much. Early adolescents selected foods for convenience, taste preferences, and availability. Foods selected during iEOs were based on parent, household and early adolescent factors, which could be addressed to influence overall diet quality.

## 1. Introduction

Eating occasions are typically experienced within a social setting based on cultural traditions [1,2]. Among early adolescents (10–13 years), eating occasions usually occur at home during family meals or at school among peers. Independent eating occasions (iEOs) where the adolescent eats meals and snacks when main caregivers are not present can be an alternative to family meals. iEOs may occur more often if the frequency of family meals decreases with shifts in family structure and routine and as the child grows older and becomes more autonomous [3,4].

Information regarding the frequency of iEOs among early adolescents is limited. In addition, available frequency data are based on measures that are not comparable. For example, approximately one-fifth (21%) of adolescents in a U.S. national sample of web-based survey respondents agreed that they often eat alone [5]. Results from other studies indicated that dinner may be the least likely meal eaten alone based on surveys with 7th grade Canadian students [6] and Japanese adolescents [7] and on diaries of activities from 15:00 to 00:00 kept by U.S. middle school students [8].

The contribution that foods and beverages consumed during iEOs make to overall diet quality and weight status of early adolescents is limited to several studies. Among Japanese adolescents, those who ate breakfast alone had a lower daily intake of fish, vegetables and fruit compared to adolescents who ate breakfast with parents/caregivers [9]. Less frequent family meals were associated with low intake of fruits and vegetables and high soft drink intake among adolescents based on Project EAT survey data [10]. Survey data from a national sample of U.S. adolescents showed that junk food and sugary drink daily intake frequency was positively associated with often eating alone as well as having a BMI categorized as overweight or obese [5]. The odds ratio for being overweight was significantly increased among Japanese adolescent girls who reported eating dinner alone compared to girls who did not report eating dinner alone [7]. Children (7–10 years) were less likely to eat vegetables when eating alone compared to eating with parents [4]. In contrast, 4 day food diaries from adolescents in the UK showed a higher likelihood of eating non-core foods, such as sugary drinks, sweets and salty snacks, when eating with family and friends compared to eating alone with no association observed between BMI and eating alone [11].

The likelihood of eating alone among early adolescents may be influenced by parental and family demographic and household characteristics that could contribute to less adult supervision. For example, more middle school students with a low school-level socioeconomic status (SES) reported that they ate dinner alone than students with a higher school-level SES in a web-based survey [6]. The 2014/1015 UK Time Survey I showed that adults in the higher occupational class spent more time eating together with family members and those with a higher level of education had more frequent family meals [12].

Characteristics of family meal and snacking environments for children and adolescents during iEOs have been identified previously using ecological momentary assessment (EMA) [13,14,15]. EMA involves repeated sampling of subjects’ current behaviors and experiences in real time, in subjects’ natural environments [16]. Eating context was studied among UK children (7–10 years) including location, others present, whether watching TV and if eaten at a table [4]. Family meal environments were also examined among a multiethnic sample of families with younger children (5–7 years) to identify the preparer, meal type and foods served, influences on what was served, rules about meals, who was present, and where the meal was served [13]. Decisions about foods served were typically based on family preferences and the preparer’s desire to serve healthy foods with a variety of options. Rules about family meals were related to use of electronics. Grenard et al. [14] found that unhealthy snack consumption among adolescents was associated with being with friends or at school, feeling lonely or craving a snack. A similar assessment regarding iEOs among early adolescents has not been conducted.

The frequency of iEOs, environmental context and relationship to eating behaviors among early adolescents have not been well characterized; however, eating alone has generally been associated with less healthy diets and overweight in most studies. Demographic factors, such as low SES, may contribute to meal and snack patterns that include more frequent iEOs for early adolescents. A better understanding of the environmental context and relationship to food intake during iEOs could inform the development of interventions for parents and early adolescents to improve the quality of foods and beverages consumed during iEOs by early adolescents. The purpose of this study was to describe iEOs among low-income early adolescents (10–13 years), including the environmental context, foods selected, reasons for choosing these foods and parental rules identified by early adolescents related to foods consumed during iEOs.

## 2. Materials and Methods

### 2.1. Participants

This formative, cross-sectional study was conducted among a convenience sample of low-income early adolescents (10–13 years, n = 51) in 10 U.S. states (AZ, CT, GA, HI, IN, KS, OH, OR, MN and UT) and the District of Columbia. Several (n = 4–5) participants were recruited from each state from community organizations serving low-income families through flyers, program announcements and word of mouth. A parent/caregiver provided permission while the early adolescent gave assent to enroll and participate in the study. Participants received cash, a gift card, or Fitbit as compensation for participation. The protocol was reviewed and approved by the Human Subject Institutional Review Boards of each of the participating institutions.

### 2.2. Data Collection

A detailed protocol was developed and used by researchers in all states to guide data collection to maintain consistency in recruitment and interview methods. Researchers had extensive experience conducting interviews regarding dietary intake. In some states, research assistants were trained to conduct the semi-structured interviews.

At Time 1, researchers met with participants to explain the study protocol. Participants were instructed to take up to three photos per eating occasion of all foods and beverages consumed on a chosen day, and to note the context within which they were consumed for the entire day. The participant was instructed to send the photos via text messaging to the researchers’ cell phones immediately after taking the photos or to bring the phone or camera to the interview at Time 2.

At Time 2, researchers interviewed the participant using the photos as a guide. Interviews took place no more than 48 h after participants completed taking photos. Researchers asked questions about each eating occasion (EO) including the type of EO (meal or non-meal snack), location, other activities taking place during the EO (e.g., watching TV), who was present (e.g., no one, mother, sibling, friend, etc.), if all present consumed the same food/beverage, the reason for choosing the food/beverage, whether other food/beverage options were available and whether parent(s) had rules or expectation about what to consume during the EO. The photos of foods/beverages consumed were ordered in a PowerPoint presentation as slides in the sequence in which they were consumed to allow the participant to better describe the context surrounding the EO, particularly the iEOs. If participants identified consuming a food(s) not shown in a photo, the interviewer added a blank slide in the appropriate place and manually added details about the food/beverage consumed.

### 2.3. Defining Variables

#### 2.3.1. iEOs

EO were considered iEOs if the participant responded “Yes” to the question “Were you eating alone?” or indicated they were eating with someone other than their mother, father, grandparent, and/or aunt. Interviews with participants at Time 2 indicated that mothers, fathers, grandparents and aunts were commonly considered primary caregivers who ate with the participant and provided food for meals/snacks. Therefore, EO where these individuals were not present were classified as iEOs.

#### 2.3.2. Liking Ratings of Foods and Expected Parental Approval

Participants were asked to indicate how much they liked the food/beverage (1 = don’t like it at all and 5 = like it a lot) and how much their parents wanted them to consume the foods/beverages in each EO (1 = your parents would not like it at all to 5 = they would like it a lot).

#### 2.3.3. Reasons for Participant Choosing Foods

Researchers asked participants an open-ended question to determine the reasons for choosing foods/beverages: “Tell me the reasons why you chose this food/beverage,” with a follow-up probe, “were there any other reasons?”.

#### 2.3.4. Parental Rules

Researchers asked participants the following questions to determine whether parents had rules surrounding foods consumed at iEOs: “Do your parents have any rules/expectations about eating these foods or beverages?” Responses of “yes” prompted a follow-up question: “Which foods and what are the rules?” When appropriate, researchers changed the term “parent” to the primary caregiver preparing food for the child (e.g., a grandma).

### 2.4. Data Analysis

Descriptive statistics were used to report means and frequencies for most variables.

Two researchers trained in qualitative research methods (RR, BJ) independently reviewed the list of foods consumed across the iEOs experienced by participants and coded the foods into ten groups, based on the United States Department of Agriculture’s MyPlate food groups and previous research [17,18,19,20,21,22] (Table 1). Researchers discussed any discrepancies in coding and reconciled differences. Food photos were reviewed if the researchers had any questions about the details of the foods consumed. For example, if “hoagie sandwich” was recorded from the data collection sheet, researchers reviewed the food photo to see if any additional food toppings could be identified. Seven foods were classified as “unknown” because researchers were not able to identify the food or beverage from the photo image or no descriptive data (written text) was available.

The number and percent of total iEOs across the sample in which at least one food was consumed from each of the ten food categories was calculated. For example, at least one food from the “Sweets” category was consumed at 50/176 iEOs (28.4%). Means and standard deviations were also calculated to provide a mean number of times a food category was consumed across all iEOs. If a child consumed more than one type of food from a food category at one iEO, this was counted independently. For example, if strawberries and mango were consumed at one iEO, a “2” was assigned for the “Whole Fruits” food category.

The same two trained researchers independently coded data reported by youth related to the reasons they chose the foods at each iEO and any parental rules related to foods consumed at iEOs. Researchers met to discuss initial codes and collapsed them into a more concise set of codes that were agreed upon by both researchers. These new sets of codes were used to re-code the data independently. Researchers held a series of meetings to review any discrepancies in coding and reconciled differences. The number and percent of youth reporting each identified parental rule was calculated; most participants reported more than one parental rule at iEOs, thus percentages calculated across identified rules do not total 100%. The number and percent of each reason given for food selection was calculated across all iEOs.

Some sites had participants provide a liking rating regarding liking foods for the overall eating occasion (*n* = 15, 29%) and others had the child report a liking rating for each food in the eating occasion (*n* = 25, 49%). To make the values comparable, for the sites that asked for a rating for each food, a pooled average was calculated for each iEOs. Ratings for six participants (*n* = 12%) were not available for any foods consumed during iEOs and five participants (10%) did not consume iEOs, thus these were excluded from analyses. Ratings were also missing for foods consumed at one iEO for seven participants; however, ratings were available for these participants for foods consumed at other iEOs in the day, thus pooled averages across iEOs were calculated. Analyses of descriptive statistics were performed in IBM SPSS Statistics (v. 24, Armonk, NY, USA).

## 3. Results

### 3.1. Demographic and EO Characteristics/Context

Sixty percent of participants were female, and most were between 10 and 11 years of age (Table 2). Youth identified 304 total eating occasions, with 57.9% classified as iEOs (data not shown). Five participants did not identify any iEOs. The majority of iEOs were consumed at home, were self-prepared, and were considered a snack (Table 3). The majority of youth reported only eating during iEOs (not combined with any other activity) or watching/playing on some form of media (Table 3).

### 3.2. Foods Consumed and Reasons for Food Selection

The top four most frequently consumed foods at iEOs were within the food categories of sweets, total fruit, dairy, and whole fruit (Table 4). Non-sugary cereals were the least frequently consumed food category at iEOs.

The top three reasons reported for choosing foods across all iEOs were food preference (taste/like it) at 45.5%, the only option at 9.7%, and someone made it for me (gave it to me) at 9.7% (data not shown). The least common reason reported for choosing foods across all iEOs was that the food was healthy at 3.0% (data not shown). Participants reported food preference (taste/like it) as a reason for selecting a food at iEOs for all food categories (Table 5). For all food categories, except non-sugary cereals, participants selected the food because it was the only option available. Convenience (quick/easy) was a reason given for 8 out of 10 food categories. Participants also chose foods from 8 out of 10 food categories because someone else made it for them or gave it to them and because someone else told them to eat the food. Healthy was a reason given for 6 out of 10 food categories (sweets, total fruit, whole fruit, dairy, vegetables, non-sugary cereals). Participants reported being hungry/thirsty as a reason for selecting foods from 6 out of 10 food categories (sweets, total fruit, whole fruit, dairy, vegetables and sugary cereals). Participants reported availability as a reason for choosing foods from half of the food categories (sweets, dairy, vegetables, salty snacks, and sugary cereals). Wanting to try a new food or choosing a food not consumed in a while were reasons participants reported for half of the food categories (sweets, total fruit, whole fruit, vegetables, and non-dairy SSBs).

### 3.3. Parental Rules

Thirty participants (58.8%) reported that parents had rules about foods consumed during iEOs. Among participants who reported rules at iEOs, 56.7% reported that parents did not want them to eat too much; 46.7% reported rules of avoiding/limiting certain kinds of foods, such as sweets; 13.3% of youth reported rules related to not making a mess, focusing on health with foods consumed or how food was eaten (e.g., not too fast), and saving food for others; and 3.3% reported needing to ask parents first before eating the food (data not shown).

## 4. Discussion

Early adolescents in the current study frequently made independent food choices. Most iEOs occurred at home and were considered “snacks”. One-third of participants reported parents had rules about foods eaten during iEOs, and participants often chose foods that were “quick and easy”. Findings provide valuable information to guide interventions seeking to promote healthy intake during iEOs. Data based on nationally representative or other large population-based samples of adolescents have characterized dietary intake on a daily or usual basis, while fewer studies have examined intake during specific occasions by the presence of others at the occasion and the context in which decisions are made about what to eat and drink. This formative study provides an initial, detailed examination of iEOs experienced by a diverse sample of low-income early adolescents (10–13 years), thus contributing to knowledge of how parents, household characteristics, and early adolescents themselves influence food choices during iEOs.

Many foods consumed during iEOs were energy-dense snacks eaten at home, with some containing few micronutrients. These findings align with previous studies examining adolescent snacking showing that snacks consumed were often of poor nutritional quality [23,24,25], though studies identified did not specifically focus on iEOs. One important area of focus may be the food that is made available at home for early adolescents when they are alone; which relates to food parenting practices. Of participants in the current study who reported parental rules, only approximately one-quarter mentioned parents limiting certain kinds of foods. Rules related to controlling availability of particular foods to adolescents have been found to promote healthier eating behaviors in several previous studies [26,27,28,29].

In addition, early adolescents in the current study reported a preference for quick and easy foods they liked as the primary reasons for choosing foods during iEOs. Given that many iEOs were considered a snack, this preference may be particularly relevant to these eating occasions. Promoting snacks that are quick, easy, and healthy should be an area of focus for nutrition educators. This was the focus of a study performed with African-American school-aged girls in a community setting that involved teaching participants how to prepare easy recipes that may be consumed during various meals and snacks [30]. For example, girls learned to prepare fruit smoothies, yogurt parfaits, bean-cheese tacos and turkey wraps. Parents may potentially promote intake of such snack foods when they are not around by making simple ingredients available so that adolescents may use them in recipes or preparing items in advance for easy access.

In the current study, reasons for selecting foods for consumption during iEOs were reported during individual retrospective interviews based on photos of foods consumed over a 24 h period. Older adolescents (14–17 years) in another study indicated via an ecological momentary assessment protocol over 7 days that factors such as feelings of loneliness or boredom, cravings and exposure to cues promoted unhealthy snack selection in general [14]. Differences in methodology, type of eating occasions, and level of sophistication regarding ability to verbalize reasons for food choices among older vs. younger adolescents may explain differences in reasons given for food choices. Early adolescents may also have been influenced by similar reasons for food choices during iEOs as older adolescents but may not have been able to recall specific motivations during a retrospective discussion. Future studies could utilize similar EMA protocols with younger adolescents to better capture motivation for food choices during iEOs.

Approximately two-thirds of the foods consumed in the current study during iEOs were considered snacks and approximately one-third were considered a meal, with approximately about half of the participants indicating that they prepared the foods they consumed during iEOs. These findings highlight the need for early adolescents to have adequate food preparation skills and nutrition knowledge to make healthy choices during iEOs. However, focus group discussions with Irish mothers indicated a lack a transference of cooking skills with limited opportunities for children to learn basic skills [31]. Data collected from a wearable all-day camera indicated that early adolescents did little actual food preparation; most activities involved simply opening packages and combining several ingredients [32]. Because youth may be responsible for selecting and preparing foods for iEOs, future interventions could focus on helping parents teach children food preparation skills and benefits of healthy eating based on healthy portions to support autonomy during iEOs.

Early adolescents identified parental rules in the current study largely related to preventing children from eating too much and limiting certain kinds of foods. Numerous studies have demonstrated that creation and enforcement of food-related rules at home are promising parenting practices intended to promote intake of healthy foods and restrict intake of unhealthy foods [33,34,35]. A previous study among diverse adolescents in California showed that adolescents reporting at least one health-oriented food rule at home were significantly more likely to make healthy independent snack choices than those reporting no rules [36]. Adolescents with health-oriented rules at home were also more likely to indicate they felt guilty when choosing unhealthy food and good when choosing healthy foods for snacks than those without rules. The authors noted that food rules may shape youth beliefs and perceptions with a clearer understanding of boundaries and guidelines for eating, allowing parents to guide adolescents to make healthier food choices on their own. Future studies may investigate specific parental rules and effects on early adolescent eating behavior during iEOs.

Of note, participants in the current study were diverse early adolescents from low-income families. In reviewing the findings, a host of factors influencing eating choices in this group need to be considered. A previous systematic review, for example, revealed racial/ethnic differences in fruit and vegetable intake and pointed to the need to consider various influences on intake when designing dietary interventions for low-income youth [37]. Another study in low-income African-American adolescents showed that specific social relationships have specific roles; for example, aunts provided exposure to novel food experiences [38]. In conducting future studies in this demographic, examination of all the contextual factors shaping adolescent eating will be important.

Limitations of this study include use of retrospective, self-reported data from a small convenience sample of early adolescents, which limited the ability to make comparisons by sex of participants. However, the use of photos of the foods consumed during all eating occasions to guide the discussion likely strengthened the ability to recall the situational context and motivation for selecting foods. Other observational methods including EMA, that capture eating behaviors and reactions as they occur, may be useful in collecting future data regarding iEOs from early adolescents. A strength of the study was the collection of data from low-income, multiethnic, youth, thus providing a diverse sample upon which to draw conclusions regarding iEO environmental or situational contexts. In addition, the results showed that the majority of EOs were independent, thereby providing justification for describing foods selected and the environmental context.

## 5. Conclusions

Early adolescents frequently made independent food choices. The majority of the iEOs were snack occasions taking place at home with foods that were self-prepared. Common reasons for food selection during iEOs were convenience, preferences and availability. Parental rules about foods consumed during iEOs were intended to prevent youth from eating too much and to restrict certain foods. Understanding choices made within the environmental context may allow parents to promote healthy eating habits among youth in this age group through positive food parenting practices.

## Figures and Tables

**Table 1 ijerph-17-00981-t001:** Types of foods consumed at independent eating occasions (iEOs) among early adolescents aged 10–13 years ^1^.

Food Category	Types of Foods
Total fruit	Oranges (including mandarin); juice (orange, cranberry, apple, 100% juice Capri-sun); grapes; pomegranate; mango; Gogo-squeeze applesauce; banana; cantaloupe; apple; pineapple; pears; strawberries
Whole fruit	Orange (including mandarin); grapes; pomegranate; mango; Gogo-squeeze applesauce; banana; cantaloupe; apple; pineapple; pears; strawberries
Vegetables	Tomato/marina sauce (on spaghetti); pizza sauce on pizza; carrots; tomato; beans/legumes (including bean dip); broccoli; lettuce; corn; sub sandwich w/vegetables; potatoes (sweet potatoes, potato salad, mashed); cabbage; okra soup; V8
Dairy	Cheese (or foods made with cheese including macaroni and cheese, alfredo, tacos, pizza, sandwich, bean and cheese burrito, egg and cheese bagel); white milk; yogurt; M&M YoCrunch
Dairy-SSBs ^2^	Chocolate milk; vanilla strawberry Nesquick; ice cream milk shake
Non-dairy SSBs	Sports drinks (Gatorade; Powerade); punch (Hi-C, Pog passion fruit juice, orange drink, Aloe Vera King juice); soda; sweet tea
Sugary cereals	Froot Loops; Cinnamon Toast Crunch; Frosted Flake; Honey Bunches of Oats; Cookie Crisps; Captain Crunch; sugar cereal w/marshmallows; Cocoa Pebbles; Apple Cinnamon Cheerios; Mini-wheats; Corn Pops; Rice Krispies Holiday Cereal
Non-sugary cereals	Oatmeal; original Cheerios
Sweets	Ice cream; sweetened grains (pop-tart, funnel cake, chocolate-covered pretzels, corn bread, muffins, donut, granola bar); candy (M&Ms, Mentos, Lifesavers, Starburst, Twix, Tootsie rolls, sucker, push popu); pie; marshmallows; cookies/bars; brownie; cake; trail mix bar; Fruit by the Foot; fruit-flavored snacks; chocolate syrup; jelly
Salty snacks	Crackers (CheezIt, round butter crackers); pizza bites/rolls/bagels; chips (Pringles, Doritos, barbecue chips, tortilla chips, Hot fries, Hot Cheetos, Funyuns, Lays Stax); popcorn; pretzels; tator tots

^1^ Independent eating occasions were identified by participants responding “Yes” to the question “Were you eating alone?” or if they indicated they were eating with anyone other than their mother, father, grandparent, and/or aunt. ^2^ SSBs = Sugar-sweetened beverages.

**Table 2 ijerph-17-00981-t002:** Demographic characteristics of participants (10–13 years).

Characteristic	*n* (%)
Sex	
Male	21 (41.2)
Female	30 (58.8)
Age (in years)	
10	24 (47.1)
11	11 (21.6)
12	7 (13.7)
13	9 (17.6)
Age, mean (SD)	11.0 (1.2)
Hispanic/Latino	
Yes	20 (39.2)
Race ^1^	
African American	14 (28.6)
Asian	2 (4.1)
Pacific Islander	2 (4.1)
White/Caucasian	15 (30.6)
Other (multiracial, multiethnic, or undefined)	16 (32.7)

^1^ Missing data, *n* = 2.

**Table 3 ijerph-17-00981-t003:** General description of independent eating occasions (iEOs) among participants (10–13 years) ^1^.

Characteristic	*n* (%)
Eating occasion type	
Meal	60 (34.1)
Snack	116 (65.9)
Specific eating location ^2^	
My home ^3^	127 (72.6)
School	32 (18.3)
At someone else’s house	8 (4.6)
In car or bus	2 (1.1)
Other location	6 (3.4)
Other activity while eating ^4^	
Just eating	60 (34.5)
Watching TV/YouTube/playing video games/surfing internet	58 (33.3)
Hanging out with a friend	27 (15.5)
Studying/reading	2 (1.1)
At an after-school program/lessons/other	2 (1.1)
Other	25 (14.4)
Self-prepared ^5^	
Yes	82 (47.4)
Partially	1 (0.6)
No	90 (52.0)
	Mean (SD)
Liking of foods consumed at iEOs ^6,7^	4.2 (1.0)
Perception of parents’ desire for participant to eat foods consumed at iEOs ^8,9^	3.7 (1.2)

^1^ Independent eating occasions were identified by participants responding “Yes” to the question “Were you eating alone?” or if they indicated they were eating with anyone other than their mother, father, grandparent, and/or aunt. ^2^ Missing data, *n* = 1 iEOs. ^3^ Seventeen (13.4%) were consumed in participant’s room. ^4^ Missing data, *n* = 2 iEOs. ^5^ Missing data, *n* = 3 iEOs. ^6^ Missing data, *n* = 14 iEOs. ^7^ How much do you like each food/beverage? (1 = don’t like at all, 5 = like a lot). ^8^ Missing data, *n* = 16 iEOs. ^9^ How much would your parents want you to eat this food? (1 = your parents would not like it at all, 5 = they would like it a lot).

**Table 4 ijerph-17-00981-t004:** Frequency of foods consumed by food category during independent eating occasions (iEOs) among participants (10–13 years) ^1^.

Food Category	Frequency of iEOs	
	n (%) ^2^	Mean (SD) ^3^
Sweets	50 (28.4)	0.30 (0.48)
Total fruit	42 (23.9)	0.27 (0.53)
Dairy	35 (19.9)	0.22 (0.48)
Whole fruit	34 (19.3)	0.22 (0.49)
Vegetables	22 (12.5)	0.16 (0.45)
Salty snacks	25 (14.2)	0.15 (0.37)
Non-dairy SSBs ^4^	14 (8.0)	0.08 (0.27)
Sugary cereals	14 (8.0)	0.08 (0.27)
Dairy SSBs	5 (2.8)	0.03 (0.17)
Non-sugary cereals	3 (1.7)	0.02 (0.13)

^1^ Independent eating occasions were identified by participants responding “Yes” to the question “Were you eating alone?” or if they indicated they were eating with anyone other than their mother, father, grandparent, and/or aunt. ^2^ Number and percent of total iEOs in which at least one food was consumed from that category. For example, at least one food in the “Sweets” category was consumed at 50/176 iEOs = 28.4%. ^3^ Mean number of times that food category was consumed across all iEOs of participants (10–13 years). ^4^ SSBs = Sugar-sweetened beverages.

**Table 5 ijerph-17-00981-t005:** Reasons for choosing foods at independent eating occasions (iEOs) by food category among participants (10–13 years) ^1^.

**Food Category**	**Food Preference (Taste/Like It)**	**Only Option Available**	**Convenience (Quick/Easy)**		**Someone Else Made It for Me (Gave It to Me)**		**Someone Else Told Me to Eat It**
Sweets	X	X	X		X		X
Salty snacks	X	X	X		X		X
Total fruit	X	X	X		X		X
Whole fruit	X	X	X		X		X
Vegetables	X	X	X		X		x
Dairy	X	X	X		X		X
Dairy SSBs ^2^	X	X					
Non-dairy SSBs ^2^	X	X			X		
Sugary cereals	X	X	X				X
Non-sugary cereals	X		X		X		
**Food Category**	**Healthy**	**Hungry/Thirsty**		**Availability**		**Wanting to Try a New Food or Choosing a Food Not Consumed in a While**	
Sweets	X	X		X		X	
Salty snacks				X			
Total fruit	X	X				X	
Whole fruit	X	X				X	
Vegetables	X	X		X		X	
Dairy	X	X		X			
Dairy SSBs ^2^							
Non-dairy SSBs ^2^						X	
Sugary cereals		X		X			
Non-sugary cereals							

^1^ Independent eating occasions were identified by participants responding “Yes” to the question “Were you eating alone?” or if they indicated they were eating with anyone other than their mother, father, grandparent, and/or aunt. ^2^ SSBs = Sugar-sweetened beverages.

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
