# Peer review of "Describing Independent Eating Occasions among Low-Income Adolescents"

_ijerph, 2020, doi:10.3390/ijerph17030981_

Round 1
Reviewer 1 Report
Reviewer comments
This research study examined independent eating occasions among low-income early adolescents including the examination of what the adolescents ate, the environmental context, reasons for choosing foods, and parental rules. The document is well-written, and the study is timely and interesting. It is worthy of publication, but I do have a few suggestions to improve the manuscript as described below.
Materials and Methods
You mentioned that adolescents lived in one of 10 states and that 46 were interviewed, but how many came from each of those 10 states? Also, you mentioned that a detailed protocol was followed, but did one person in each state collect the data or did investigators travel around to each of the 10 states to collect data? If multiple researchers collected data, how were they trained (conference call? Site visit?)? Were all the data sent to one site for analysis?
You mention that “If a missing food/beverage or EO occurred, the interviewer added a blank slide. . .” How would they have known that it was missing if there was no picture? Did the child tell them that they forgot to take a picture? Please clarify this statement.
You mention in the paragraph before the Results section that “Some” sites had early adolescents provide a liking rating for the overall eating occasions and others had the child report liking ratings for each food in the eating occasion. Please provide the numbers and percentages.
Results
I recommend that you present your results in the order that they were mentioned in your purpose statement or revised your purpose statement to read in the order that you reported your results. For example, your purpose statement indicates that you will describe the 1) environmental context, 2) foods selected, 3) reasons for choosing these foods, and 4) parental rules identified by early adolescents, but the first results you present (after the children’s demographic) are “parental rules.” It doesn’t matter about the order, just be consistent.
3.1 Demographics and EO characteristics
The table mentioned in the first sentence of the first paragraph should be Table 2. The table mentioned in the fourth sentence should be Table 3
Table 2
Do not present the data for “No” for Hispanic/Latino. Just show the “Yes” information.
Table 3
Footnote #6: Is this stated correctly that five children didn’t have any iEOs? The footnote is referring to something about parent rules, so why does the statement refer to iEOs? I am confused.
Discussion
The first sentence in the second paragraph mentions that “energy-dense snacks eaten at home had “little nutritional value.” This is too strong a statement. Please re-phrase it.
In the fourth sentence in the third paragraph, please state that the study was performed “with” not “in” African American school-aged girls. . .
In the fourth paragraph you mention “ecological momentary assessment.” Please provide a brief explanation of this type of data gathering, as the reader may not be familiar with the method.
Limitations
It is not clear to me as to who collected the data at each of the 10 sites, but if multiple persons collected data, then you should mention this as a limitation.
Author Response
Reviewer #1
Materials and Methods
You mentioned that adolescents lived in one of 10 states and that 46 were interviewed, but how many came from each of those 10 states? Also, you mentioned that a detailed protocol was followed, but did one person in each state collect the data or did investigators travel around to each of the 10 states to collect data? If multiple researchers collected data, how were they trained (conference call? Site visit?)? Were all the data sent to one site for analysis?
One investigator in each state collected the data from 4-5 early adolescents.
We revised the methods as follows to provide this information:
Participants section:
Several (n = 4-5) participants were recruited from each state from community organizations serving low-income families through flyers, program announcements and word of mouth.
Investigators all have extensive experience in collecting dietary data. Investigators meet once per year to develop and discuss research protocols, and then communicate once per month and as needed about ongoing research. Investigators collectively developed the protocol and had the required expertise for data collection. A sub-group of investigators completed the analysis and communicated remotely to do so.
Data collection section:
A detailed protocol was developed and used by researchers in all states to guide data collection to maintain consistency in recruitment and interview methods. Researchers had extensive experience conducting interviews regarding dietary intake. In some states, research assistants were trained to conduct the semi-structured interviews.
You mention that “If a missing food/beverage or EO occurred, the interviewer added a blank slide. . .” How would they have known that it was missing if there was no picture? Did the child tell them that they forgot to take a picture? Please clarify this statement.
Yes, participants were asked if there were any foods missing, and if they reported this, a blank slide was added.
We revised the methods to better clarify this statement:
Data Collection section:
If participants identified consuming a food/beverage not shown in a photo, the interviewer added a blank slide in the appropriate place and manually added details about the food/beverage consumed.
You mention in the paragraph before the Results section that “Some” sites had early adolescents provide a liking rating for the overall eating occasions and others had the child report liking ratings for each food in the eating occasion. Please provide the numbers and percentages.
We added numbers and percentages to this section.
Data Analysis section:
Some sites had participants provide a liking rating regarding liking foods for the overall eating occasion (n=15, 29%) and others had the child report a liking rating for each food in the eating occasion (n=25, 49%). To make the values comparable, for the sites that asked for a rating for each food, a pooled average was calculated for each iEOs. Ratings for six participants (n=12%) were not available for any foods consumed during iEOs and five participants (10%) did not consume iEOs, thus these were excluded from analyses. Ratings were also missing for foods consumed at one iEO for seven participants; however, ratings were available for these participants for foods consumed at other iEOs in the day, thus pooled averages across iEOs were calculated.
Results
I recommend that you present your results in the order that they were mentioned in your purpose statement or revised your purpose statement to read in the order that you reported your results. For example, your purpose statement indicates that you will describe the 1) environmental context, 2) foods selected, 3) reasons for choosing these foods, and 4) parental rules identified by early adolescents, but the first results you present (after the children’s demographic) are “parental rules.” It doesn’t matter about the order, just be consistent.
3.1 Demographics and EO characteristics
The table mentioned in the first sentence of the first paragraph should be Table 2. The table mentioned in the fourth sentence should be Table 3
We revised these sentences based on the reordered results section.
Table 2
Do not present the data for “No” for Hispanic/Latino. Just show the “Yes” information.
Table 2 was revised to remove the “No” information.
Table 3
Footnote #6: Is this stated correctly that five children didn’t have any iEOs? The footnote is referring to something about parent rules, so why does the statement refer to iEOs? I am confused.
We removed this statistic from Table 3 and instead described it in the text only, given it was referring to number of participants reporting iEOs and the rest of the table was about iEOs, not participants. We found an error in the data described, which we fixed in this revision. The new text added to the Results reads as follows:
Thirty participants (58.8%) reported that parents had rules about foods consumed during iEOs. Among participants who reported rules at iEOs, 56.7% reported that parents did not want them to eat too much; 46.7% reported rules of avoiding/limiting certain kinds of foods, such as sweets; 13.3% of youth reported rules related to not making a mess, focusing on health with foods consumed or how food was eaten (e.g. not too fast), and saving food for others; and 3.3% reported needing to ask parents first before eating the food (data not shown).
Discussion
The first sentence in the second paragraph mentions that “energy-dense snacks eaten at home had “little nutritional value.” This is too strong a statement. Please re-phrase it.
See the first sentence in the second paragraph of the Discussion section. We have re-phrased this sentence to read: Many foods consumed during iEOs were energy-dense snacks eaten at home, with some containing few micronutrients.
In the fourth sentence in the third paragraph, please state that the study was performed “with” not “in” African American school-aged girls. . .
We have updated this. The sentence now reads: This was the focus of a study performed with African-American school-aged girls in a community setting that involved teaching participants how to prepare easy recipes that may be consumed during various meals and snacks [29].
In the fourth paragraph you mention “ecological momentary assessment.” Please provide a brief explanation of this type of data gathering, as the reader may not be familiar with the method.
We have added this explanation to the Introduction, page 2: EMA involves repeated sampling of subjects' current behaviors and experiences in real time, in subjects' natural environments.
Limitations
It is not clear to me as to who collected the data at each of the 10 sites, but if multiple persons collected data, then you should mention this as a limitation.
We thank the reviewer for this suggestion. However, we would not deem this to be a limitation of the study, as all investigators used the same protocol for data collection and were trained on study procedures. There was no negative effect of this on data quality.
Reviewer 2 Report
Manuscript review IJERPH 7120960
Strengths include a focus on low-income early adolescents from diverse regions of the U.S. (and suggested differences between this age group and older adolescents in the Discussion); data collection methodology (participants’ photos of all food eaten and interview with researcher within 48 hours); the finding that the majority of eating occasions among study participants are independent, justifying the work describing the content of those occasions; and mention of potential applications of the results.
Weaknesses include a small sample size; limited data from each participant (eating during a single day); lack of a gender analysis when previous literature suggests that girls and boys make different eating choices; lack of rationale for the definition of independent eating occasions; and the fact that this study is simply descriptive, with means and frequencies the only statistics. With so many authors, is this study part of a larger study? Can the findings reported here be included in the write-up of another aspect or aspects of a larger project?
Additional comments:
Section 2.1
Too much procedure described in the Participants section. Remove sentences 3-5 from this section.
Mean age should be included in this section. (“Sixty percent of participants were female, and most were between 10-11 years of age” and Table 2 containing demographic descriptors appear in section 3.1 but should appear in 2.1 and be considered descriptors of the sample rather than results of the study)
Did participants receive compensation?
Section 2.2à2.3
Unclear how data were quantified to enable the descriptive statistics.
Likert-type coding for “Liking” and “expected parent approval” variables is explained, but not for the other variables
Maybe separate out a section on “Variables” from section 2.2 to explain how each variable was measured. Such a “Variables” section could also explain why eating with others (besides parents, grandparents, and aunts), in addition to eating alone, is considered an IEO. What is it about parents, grandparents, and aunts that creates a fundamentally different eating occasion for the young adolescent?
Section 3.3 Para 2 Line 1-2, Line 4 & Table 5
Inconsistent terminology leads to some confusion on why participants were choosing certain foods. Text at line 2 indicates that “food preference” is the same as “quick and easy” whereas line 4 describes “preference” as “taste/like” and line 6 connects “quick/easy” with “convenience” and the table conflates “preference” with “liking” (the “prefer/like” column) and has separate categories for “convenient” and “available” which sound more like “quick and easy.” Please use consistent terms in both text and table.
Grammar/Writing
Why is only the “i” in iEO (“independent eating occasions”) capitalized? I see the authors have done this in their published work elsewhere, but not in their poster of the current study in which the I is capitalized.
Probably don’t need to refer to participants as “early adolescents” every time. It gets awkward. A few times the word “youth” is substituted when referring to the participants. Just the word “participants” in all instances would be more familiar and flow more smoothly.
For example, in labels for Tables 2-5 -- it would be clearer that you’re describing your sample if “early adolescents” here was replaced with the word “participants” or “sample”
Table 1
Some inconsistent capitalization. For example, the following words should be lower-case to remain consistent with other lower-case items: juice, punch, tea, sugar, barbecue, chips, tortilla
Section 3.1 Line 1:
“…most were between 10-11 years of age …” “between A to B” is ungrammatical. Should be either “between A and B” (preferred) or “A to B” (i.e., A-B)
Table 2 is mentioned as showing “The majority of youth reported only eating during iEOs (not combined with any other activity) or watching/playing on some form of media” but Table 2 does not address this finding, only demographic descriptors of the sample
Section 3.2 Line 5:
Grammar: three instances of number disagreement:
p 2 para 1line 3
“The alternative to family meals include…” should be “An alternative to family meals is…” (or at least fix number agreement to “includes”)
p 2 para 2 line 1
“The contribution that foods and beverages consumed during iEOs makes to overall diet quality…” “makes” should be “make”
Table 1 caption
“1Independent eating occasion were identified…” should be “occasions” or “was”
Author Response
Reviewer #2
Strengths include a focus on low-income early adolescents from diverse regions of the U.S. (and suggested differences between this age group and older adolescents in the Discussion); data collection methodology (participants’ photos of all food eaten and interview with researcher within 48 hours); the finding that the majority of eating occasions among study participants are independent, justifying the work describing the content of those occasions; and mention of potential applications of the results.
We thank the reviewer for identifying these strengths.
Weaknesses include a small sample size; limited data from each participant (eating during a single day); lack of a gender analysis when previous literature suggests that girls and boys make different eating choices; lack of rationale for the definition of independent eating occasions; and the fact that this study is simply descriptive, with means and frequencies the only statistics. With so many authors, is this study part of a larger study? Can the findings reported here be included in the write-up of another aspect or aspects of a larger project?
We thank the reviewer for these comments. This study was conducted as part of an ongoing multi-state collaboration. We have chosen to report these findings as a stand-alone report given that these data are distinct from others collected as part of our work and make a unique contribution to the literature on their own.
The limitations section was expanded to indicate the use of a small convenience sample which limited the ability to make comparisons by participant sex. We also added a statement to indicate: In addition, the results showed that the majority of EOs were independent, thereby providing justification for describing foods selected and the environmental context.
Additional comments:
Section 2.1
Too much procedure described in the Participants section. Remove sentences 3-5 from this section.
We removed sentences 3-5 from the Participants section. We included some of this information in the Data collection section as follows:
A detailed protocol was developed and used by researchers in all states to guide data collection to maintain consistency in recruitment and interview methods. Researchers had extensive experience conducting interviews regarding dietary intake. In some states, research assistants were trained to conduct the semi-structured interviews.
Mean age should be included in this section. (“Sixty percent of participants were female, and most were between 10-11 years of age” and Table 2 containing demographic descriptors appear in section 3.1 but should appear in 2.1 and be considered descriptors of the sample rather than results of the study)
We added participants’ mean (SD) age to Table 2, 11.0 (1.2). We prefer to maintain the original organization as it more conventional and follows many journal standards. Therefore, information describing specific demographic characteristics in detail remains in the results rather than methods section.
Did participants receive compensation?
Yes, participants received a gift card as compensation. The following added to the Participant section:
Participants received cash, a gift card, or Fitbit as compensation for participation.
Section 2.2à2.3
Unclear how data were quantified to enable the descriptive statistics.
Likert-type coding for “Liking” and “expected parent approval” variables is explained, but not for the other variables
Maybe separate out a section on “Variables” from section 2.2 to explain how each variable was measured. Such a “Variables” section could also explain why eating with others (besides parents, grandparents, and aunts), in addition to eating alone, is considered an IEO. What is it about parents, grandparents, and aunts that creates a fundamentally different eating occasion for the young adolescent?
We created a new section, 2.3 Defining Variables and included how we defined the following variables: iEOs; Liking rating of foods and expected parental approval; Reasons for participant choosing foods; and Parental rules.
Interviews with participants at Time 2 indicated that mothers, fathers, grandparents and aunts were commonly considered primary caregivers who ate with the participant and provided food for meals/snacks. Therefore, EO where these individuals were not present were classified as iEOs.
Section 3.3 Para 2 Line 1-2, Line 4 & Table 5
Inconsistent terminology leads to some confusion on why participants were choosing certain foods. Text at line 2 indicates that “food preference” is the same as “quick and easy” whereas line 4 describes “preference” as “taste/like” and line 6 connects “quick/easy” with “convenience” and the table conflates “preference” with “liking” (the “prefer/like” column) and has separate categories for “convenient” and “available” which sound more like “quick and easy.” Please use consistent terms in both text and table.
Thank you for this feedback. We agree that this terminology does appear confusing; this was an error on our part in reporting the coding categories. We revised the text for clarification and the associated Table 5 columns now align with the terminology used in the text.
Grammar/Writing
Why is only the “i” in iEO (“independent eating occasions”) capitalized? I see the authors have done this in their published work elsewhere, but not in their poster of the current study in which the I is capitalized.
We would like to maintain consistency with our abbreviation for iEOs in our two published papers and consider the use of a different abbreviation (IEOs) in the poster as an oversight. We plan to continue using the iEO abbreviation in our future abstracts and papers.
Probably don’t need to refer to participants as “early adolescents” every time. It gets awkward. A few times the word “youth” is substituted when referring to the participants. Just the word “participants” in all instances would be more familiar and flow more smoothly.
We revised to substitute participants for early adolescents in most of the text.
For example, in labels for Tables 2-5 -- it would be clearer that you’re describing your sample if “early adolescents” here was replaced with the word “participants” or “sample”
We revised the labels for Tables 2-5 to substitute participants for early adolescents.
Table 1
Some inconsistent capitalization. For example, the following words should be lower-case to remain consistent with other lower-case items: juice, punch, tea, sugar, barbecue, chips, tortilla
The text was revised in table 1 so the words mentioned were in lower case to maintain consistency throughout.
Section 3.1 Line 1:
“…most were between 10-11 years of age …” “between A to B” is ungrammatical. Should be either “between A and B” (preferred) or “A to B” (i.e., A-B)
The sentence was revised as follows:
Sixty percent of participants were female, and most were between 10 and 11 years of age (Table 2).
Table 2 is mentioned as showing “The majority of youth reported only eating during iEOs (not combined with any other activity) or watching/playing on some form of media” but Table 2 does not address this finding, only demographic descriptors of the sample.
The table numbers were revised to match the text after reordering the results.
Section 3.2 Line 5:
Grammar: three instances of number disagreement:
We appreciate your careful review and for catching this error. The number of participants reporting parental rules around iEOs was incorrect. This has been fixed in the text, as follows:
Thirty participants (58.8%) reported that parents had rules about foods consumed during iEOs. Among participants who reported rules at iEOs, 56.7% reported that parents did not want them to eat too much; 46.7% reported rules of avoiding/limiting certain kinds of foods, such as sweets; 13.3% of youth reported rules related to not making a mess, focusing on health with foods consumed or how food was eaten (e.g. not too fast), and saving food for others; and 3.3% reported needing to ask parents first before eating the food (data not shown).
Most participants reported more than one parental rule at an independent eating occasion, thus percentages do not add up to 100%. This was clarified in the Data Analysis section:
The number and percent of youth reporting each identified parental rule was calculated; most participants reported more than one parental rule at iEOs, thus percentages calculated across identified rules do not total 100%.
p 2 para 1 line 3
“The alternative to family meals include…” should be “An alternative to family meals is…” (or at least fix number agreement to “includes”)
This sentence was revised as follows:
Independent eating occasions (iEOs) where the adolescent eats meals and snacks when main caregivers are not present can be an alternative to family meals.
p 2 para 2 line 1
“The contribution that foods and beverages consumed during iEOs makes to overall diet quality…” “makes” should be “make”
This sentence as revised as follows:
The contribution that foods and beverages consumed during iEOs make to overall diet quality and weight status of early adolescents is limited to several studies.
Table 1 caption
“1Independent eating occasion were identified…” should be “occasions” or “was”
The text was revised as follows:
1Independent eating occasions were identified by participants responding “Yes” to the question “Were you eating alone?” or if they indicated they were eating with anyone other than their mother, father, grandparent, and/or aunt.